# Overview of the Properties and Formation Process of Interface Traps in MOS and Linear Bipolar Devices

**DOI:** 10.3390/mi16040434

**Published:** 2025-04-02

**Authors:** Yanru Ren, Min Zhu, Xuehui Dai, Longxian Li, Minghui Liu

**Affiliations:** College of Nuclear Science and Technology, Naval University of Engineering, Wuhan 430033, China; d23182703@nue.edu.cn (Y.R.); min0zhu@163.com (M.Z.); xuehui1006@163.com (X.D.);

**Keywords:** transistors, radiation environment, interface trap, formation and annealing processes, research status

## Abstract

This article reviews the properties and formation process of interface traps in MOS and linear bipolar devices. Transistors are the core components of modern electronic devices, and their performance and reliability directly affect the performance of the entire system. In radiation environments, the emergence and evolution of interface traps severely impacts the functionality of transistors, being a significant factor in device failure. However, our understanding of the properties and formation processes of interface traps is still limited. Therefore, research on interface traps is of great theoretical and practical significance. This paper focuses on studying the radiation response patterns of transistor interface traps. By reviewing relevant literature and research findings from both domestic and international sources, this review provides a detailed overview of the current state of research on the transformation of interface traps and the annealing processes that occur during the irradiation of microelectronic devices. Finally, based on this foundation, this paper discusses the current state of simulation research methods for interface traps. Through an in-depth exploration of the formation mechanisms of interface traps and their role in transistor performance, this study aims to provide guidance for device design, radiation hardening, and reliability assessment, and ensure the reliability and stability of devices in radiation environments.

## 1. Introduction

Transistors are commonly used electronic components in modern electronic devices, and their performance and reliability directly affect the performance of the entire system [1]. When operating in a space radiation environment, transistors may suffer from performance degradation and failure due to interface defects caused by ionizing radiation, which can endanger the normal operation of the equipment. However, the understanding of the types and formation processes of these interface defects is still not sufficiently comprehensive, posing a research challenge for further improving the radiation resistance and reliability of devices. Therefore, research on transistor interface defects is of significant theoretical and practical importance [2]. By delving into the formation mechanisms of interface defects and their role in transistor performance, important guidance can be provided for device design, material selection, and reliability assessment, ensuring their reliability and stability in radiation environments.

This paper focuses on studying the properties of interface traps in transistors under irradiation conditions, as well as their formation and annealing mechanisms. This paper aims to provide a comprehensive analysis of the damage mechanisms and current research status of electronic devices in low-dose radiation environments, and to explore the challenges and issues in this field. By deeply understanding and addressing these problems, the paper seeks to investigate future research directions and trends in this area. This will provide a reference and guidance for better understanding the formation mechanisms of interface traps, thereby better meeting the reliability and stability requirements of electronic devices in irradiation environments and offering reliable solutions for radiation hardening [3].

## 2. The Properties and Formation Mechanism of Interface Traps

### 2.1. The Properties of Interface Traps

Interface traps are formed when hydrogen ions (released after ionizing radiation exposure generates electron–hole pairs and subsequent charge transport) migrate to the silicon–oxide interface and react with silicon–hydrogen bonds. The reaction between hydrogen and silicon–hydrogen complexes releases electrons, thereby forming interface traps [4]. Interface traps (N_it_) are defects located precisely at the Si/SiO_2_ interface. These traps have very low barriers for capturing carriers, thus greatly affecting the carrier mobility and recombination rate at the semiconductor surface [5].

Figure 1 provides a comprehensive overview of the evolution of models explaining interface traps in MOS and linear bipolar devices, particularly focusing on the enhanced low-dose-rate sensitivity (ELDRS) effect. It traces the development from the early space charge model to the more refined defect reaction competition model and the hydrogen-modified defect reaction competition model, highlighting the critical role of hydrogen in defect dynamics. The figure also references key studies and computational approaches, such as first-principles calculations, that have advanced our understanding of defect reactions and their impact on device performance under radiation. This summary encapsulates the current understanding of interface traps and sets the stage for deeper exploration in this review.

The properties of interface defects are not uniform, and both the nature of these defects and their formation processes remain poorly understood in terms of their impact on the electrical parameters of transistors. The previous literature has primarily focused on dose rates above 10 mrad (SiO_2_)/s. However, at dose rates below 1 mrad (SiO_2_)/s, the damage is observed to further increase, and the evolution of dominant radiation-induced defects at these lower dose rates remains poorly understood and lacks a precise explanation.

In 1981, Edward H. Poindexter [6] and others studied the interface states and electron spin resonance (ESR) centers in (111) and (100) silicon wafers during the thermal oxidation process. They investigated the effects of different materials and process parameters on the ESR signals and the oxide layer charge, and pointed out the potential application prospects of ESR in studying interface states and oxide layer charge. The Si/SiO_2_ interface model for (100) silicon derived from these ESR results is shown in Figure 2.

In 1989, researchers at the Center for Microelectronics Materials and Structures at Yale University, including Ma T P [7], studied the transformation characteristics of interface traps in MOS structures induced by ionizing radiation or hot electron injection. The study found that both ionizing radiation and hot electron injection can generate interface traps at the Si/SiO_2_ interface in MOS capacitors, and these traps undergo significant changes over time. After radiation or hot electron damage, the interface trap peak is located in the upper half of the Si bandgap. Over time, this peak gradually shifts to a peak in the lower half of the bandgap, forming a double-peak distribution. The study examined the interface trap behavior of MOS capacitors on Si substrates with different crystal orientations, including (100), (111), and (110). It was found that there were significant differences in the interface trap behavior of these samples after radiation. In the (111) samples, the interface trap peak gradually moved to the lower half of the bandgap, eventually forming a single peak in the lower half, while a double-peak distribution was observed in the (100) samples.

In 1989, Yu Wang [8] and colleagues from the Department of Electrical Engineering at Yale University studied the effects of radiation on the distribution of interface traps at the (111)Si/SiO_2_ interface. The study examined the behavior of interface trap distribution in metal-SiO_2_-Si capacitors after X-ray radiation, focusing specifically on silicon substrates with a (111) orientation. Immediately after radiation, the interface trap distribution in the (111) samples exhibited characteristics similar to those of (100) samples, with a significant peak appearing in the upper half of the Si bandgap. However, over time, this peak gradually shifted towards the lower half of the bandgap, eventually forming a single peak in the lower middle part of the bandgap. The study found that gate-induced compressive strain at the Si/SiO_2_ interface played a significant role in the movement of the peak position, and the presence and polarity of the gate bias also significantly affected the behavior of the interface trap distribution after radiation. The results are of great importance for understanding the behavior of interface traps in semiconductor devices under radiation conditions. Figure 3 shows a silicon tetrahedron containing dangling bond silicon defects, where d is the distance between the defect atom and the plane formed by its three nearest neighbor atoms.

In 1991, J. H. Stathis [9] and colleagues from the IBM Research Division used electron spin resonance techniques to observe a single defect (called P_b0_) at the Si(111)/SiO_2_ interface, while two different defects (called P_b0_ and P_b1_) were observed at the Si(100)/SiO_2_ interface. They found that under processing conditions where the hydrogen passivation reaction passivates the P_b_ center at the Si(111)/SiO_2_ interface, the P_b1_ center is also passivated, but the P_b0_ center is not. The study concluded that the structure of P_b1_ is similar to a silicon dangling bond like the P_b_ on (111), while P_b0_ is a fundamentally different defect. Figure 4 shows a model of the P_b_ center. At the silicon (111) interface, the P_b_ center is understood as a silicon dangling bond oriented towards the oxide layer outside the silicon wafer. At the silicon (100) interface, there are two related defects, called P_b0_ and P_b1_.

A major challenge lies in the complexity of defect dynamics, as interface traps can form through various pathways, including radiation, thermal stress, and fabrication processes. The role of hydrogen and other impurities in trap formation and passivation remains poorly understood. Additionally, accurately modeling the kinetics of trap generation and their impact on device behavior under different conditions is a significant hurdle, requiring advanced experimental and computational approaches.

In 2015, Li Xingji [10] and colleagues from Harbin Institute of Technology conducted a study on the synergistic effects of ionization and displacement defects in NPN transistors caused by 40 MeV silicon ion irradiation at low fluences. Using the 3DG110 transistor and a special gate-controlled NP-type (GNPN) transistor as samples, they investigated the synergistic effects of ionization and displacement defects through experiments with 40 MeV silicon ion irradiation. Deep-level transient spectroscopy (DLTS) was employed to characterize the defects induced by irradiation. The experimental results indicated that 40 MeV silicon ion irradiation can cause significant ionization damage signals in NP transistors (oxide trapped charges and interface traps). Through DLTS measurements, it was found that interface traps have an enhancing effect on displacement defects, while oxide trapped charges have an inhibiting effect on displacement damage. Compared to the inhibiting effect of oxide trapped charges, interface traps contribute more significantly to displacement damage.

In 2016, Chenhui Wang [11] and colleagues from the Northwest Institute of Nuclear Technology conducted experiments to explore the synergistic effects of ionization and displacement damage in gated lateral PNP bipolar transistors under mixed gamma-ray and neutron irradiation. They designed a gated lateral PNP bipolar transistor for both individual and mixed radiation experiments. The results showed that sequential irradiation with gamma rays and neutrons caused more severe degradation of the base current than individual irradiation. This is due to the positive charges and traps induced by gamma-ray irradiation in the oxide layer and at the Si/SiO_2_ interface, which enhanced the carrier recombination process in the bulk defects caused by neutron irradiation, leading to more severe degradation.

In 2019, Li Xingji [12] and colleagues from Harbin Institute of Technology studied the effects of different charged particles (10 MeV Si ions, 40 MeV Si ions, and 3 MeV protons) on the displacement damage sensitive regions in lateral PNP (LPNP) bipolar transistors. The experimental results showed that the electrical degradation of the LPNP transistor increased linearly with the increase in the radiation flux of the three types of particles. When considering the Si/SiO_2_ interface as a sensitive region, the NIEL method can be used to normalize the displacement damage caused by the three types of particles in the LPNP transistor. DLTS analysis showed that the radiation defects in the LPNP transistor caused by the three types of particles were mainly interface traps, indicating that the Si/SiO_2_ interface is a radiation-sensitive region of the device.

In 2022, Enhao Guan [13] and colleagues from Harbin Institute of Technology investigated the effects of displacement and ionization damage on interface traps in the silicon dioxide (SiO_2_) layer of bipolar transistors. In the experiment, they first pre-irradiated the GLPNP transistor with 400 keV oxygen ions to induce displacement damage in the SiO_2_ layer, followed by irradiation with Co-60 γ-rays to produce ionization damage. The concentration changes of interface traps (N_it_) and oxide traps (N_ot_) were analyzed using deep-level transient spectroscopy (DLTS) and gate sweep (GS) curves. The experimental results provided direct evidence for the space charge model, demonstrating that the oxide charges generated by displacement damage can hinder the protons produced by ionization damage from reaching the Si/SiO_2_ interface, thereby affecting the formation of interface traps.

In 2024, Binghuang Duan [14] and colleagues from the Institute of Electronic Engineering at the China Academy of Engineering Physics studied the effects of defects induced by ionizing radiation at the SiO_2_/Si interface on electronic properties, as well as the phenomenon of excess current splitting. Through low-dose-rate radiation experiments, they observed the splitting of the base current peak in gate-controlled lateral PNP (GLPNP) transistors, with this splitting varying in response to changes in radiation dose and dose rate. It was found that the current peak splitting originated from the asymmetric carrier capture cross-sections of different types of traps induced by radiation. The findings indicate that it is insufficient to describe the electronic activity of radiation-induced interface traps using a single effective carrier cross-section; the ratio of electrons to hole capture cross-sections must also be considered, as this is crucial for determining the parameters of radiation-induced interface traps. To validate the proposed theory, the authors further calculated the GS curve using TCAD simulations. Figure 5 displays the simulated response of the base current to the gate voltage. In the comparison of the simulation results in Figure 5 with the experimental results, the solid line represents the experimental data, while the dashed line and dotted line represent the TCAD simulation results and analytical model results, respectively. As the values of radiation dose increased, the I_B_ curve transitioned from a single peak to a saddle shape, resulting in two maxima whose distance continued to increase, in agreement with both the experimental and theoretical results.

### 2.2. The Mechanism of Accelerated Aging Experiments on Interface Traps

The study of interface traps in semiconductor devices under accelerated aging conditions is essential for understanding long-term reliability and performance degradation. Interface traps, arising from defects at the semiconductor–oxide interface, significantly impact electrical parameters such as threshold voltage and leakage current. Accelerated aging experiments, which simulate prolonged operational stress through elevated temperatures, high electric fields, and radiation exposure, provide insights into defect formation and evolution. However, the precise mechanisms governing interface trap generation and dynamics under these conditions remain incompletely understood, necessitating further investigation.

#### 2.2.1. The Mechanism of Accelerated Aging Experiments on Interface Traps in MOS Devices

In 1995, J.L. Titus [15] from the Naval Surface Warfare Center investigated how the thickness of the gate oxide affects the gate and drain failure threshold voltages needed to cause a single-event gate rupture (SEGR). The study uses vertical power metal-oxide semiconductor field-effect transistors (MOSFETs) with identical processes and design parameters except for the gate oxide thickness. Five different MOSFETs with nominal gate oxide thicknesses of 30, 50, 70, 100, and 150 nm were fabricated and tested with mono-energetic ion beams of nickel, bromine, iodine, and gold. By applying various bias conditions, the failure thresholds for SEGR were determined for each oxide thickness. The experimental results were used to extend a previously published empirical expression to include the effects of gate oxide thickness. Additionally, the paper briefly discusses observations related to ion angle, temperature, cell geometry, channel conductivity, and curvature at high drain voltages.

In 2006, H. J. Barnaby [16] discussed several key issues associated with deep submicron CMOS devices as well as advanced semiconductor materials in ionizing radiation environments. H. J. Barnaby introduced emerging materials and devices such as ultra-small bulk CMOS, fully depleted silicon-on-insulator (SOI), ultra-thin oxides, and high-k dielectrics, and analyzed their characteristics and challenges in the process of continuing Moore’s Law. Then, the review elaborated on the damage mechanisms of total ionizing dose effects, covering aspects such as oxide trapped charge, interface traps, and 1/f noise. Subsequently, it explored the impacts of radiation on different technologies, such as the damage of isolation oxides in ultra-small bulk CMOS and the threshold voltage drift in fully depleted SOI. Finally, it concluded that although the current prospects of electronic technologies in terms of total ionizing dose sensitivity are relatively optimistic, continuous testing and analysis of new technologies are still required to clarify their performance under extreme conditions. The paper provides rich materials for a deep understanding of the characteristics of modern CMOS technologies in radiation environments and is of great significance for research in related fields.

In 2023, Daniel M. Fleetwood [17] and colleagues at Vanderbilt University discussed the role of hydrogen in the formation and disappearance of interface traps, as well as its contribution to low-frequency noise and random telegraph noise. The migration and reaction kinetics of hydrogen are considered important factors affecting noise. The paper presented experimental results from various MOS devices, illustrating the relationship between 1/f noise and interface trap charge during post-irradiation processes. By comparing different types of MOS devices, it was found that interface traps have a more significant impact on noise than oxide traps. The activation and passivation processes of interface traps have important effects on the performance and reliability of MOS devices, especially in high-radiation environments. The paper proposed a physical model of how the activation and passivation processes of interface traps affect 1/f noise. Experimental results and first-principles calculations indicate that interface traps significantly contribute to 1/f noise in irradiated MOS devices.

In 2024, Haoshu Tan [18] and colleagues at the Microsystem and Terahertz Research Center investigated the mechanisms and effects of ionizing radiation on the dynamic characteristics of silicon carbide (SiC) metal-oxide semiconductor field-effect transistors (MOSFETs). Through experimental and theoretical analyses, the study explores the relationship between radiation-induced atomic-scale defects and the degradation of device performance, and proposes a method to predict the radiation response of MOS devices based on surface lifetime.

#### 2.2.2. The Mechanism of Accelerated Aging Experiments on Interface Traps in Linear Bipolar Devices

In 2008, Ronald L. Pease [19] and colleagues at RLP Research conducted experiments on transistors and circuits, demonstrating that hydrogen is associated with enhanced low-dose-rate sensitivity (ELDRS) in bipolar linear circuits. The experimental results showed that the amount of hydrogen determines the relationship between total dose response and dose rate, including the saturation at low dose rates and the transition dose rate between high- and low-dose-rate responses. An increase in hydrogen quantity amplified the maximum degradation at low dose rates and shifted the dose rate at which the enhancement from high to low dose rates occurs. Figure 6 presents data on the relationship between the increase in N_it_ and the dose rate after irradiation to 30 krad(Si) for the average of three samples. The data indicate that as hydrogen concentration increases, the maximum saturation degradation at low dose rates increases, and the transition between high- and low-dose-rate responses moves to higher dose rates.

In 2009, David R. Hughart [20] and colleagues from the Department of Electrical Engineering and Computer Science at Vanderbilt University explored the effects of hydrogen and aging on the radiation response of gate-controlled lateral PNP bipolar transistors. They conducted hydrogen immersion experiments to assess the defect accumulation and annealing behavior of gate-controlled lateral bipolar transistors under hydrogen exposure. They compared the radiation response of transistors tested in 2009 with those of the same devices from the same wafer tested in 2003 to study the impact of aging on radiation response. The results showed that after six years of room temperature storage, the density of radiation-induced interface traps and oxide trapped charges was lower than it was six years prior. Transistors that were hydrogen-immersed showed a higher concentration of oxide trapped charges after radiation but had a faster annealing rate. The hydrogen immersion experiments indicated that exposure to hydrogen increases the accumulation of radiation-induced interface traps and oxide trapped charges and accelerates the annealing rate of oxide trapped charges. Over time, the radiation response of hydrogen-sensitive transistors stored at room temperature improved. The role of hydrogen in the radiation response of transistors mainly depends on whether hydrogen diffuses into or out of the device and whether the initial defect concentration is conducive to net passivation or depassivation reactions with molecular hydrogen.

In 2018, XiaoLong Li [21] and colleagues from the Xinjiang Institute of Physics and Chemistry conducted a study on the low-dose-rate irradiation damage characteristics and variation patterns of four typical analog circuits using two irradiation methods: variable dose rate and variable temperature. They analyzed the changes in irradiation-sensitive parameters under both methods, compared the degradation levels of devices under different irradiation methods, and discussed the mechanisms of these two laboratory-accelerated assessment methods for low-dose-rate irradiation damage. The results indicated that the enhanced low-dose-rate irradiation damage effect in bipolar circuits is related to the density of induced interface states and hydrogenated oxygen vacancy defects. Changes in dose rate and temperature during irradiation promote the growth of interface states and inhibit their passivation, thereby activating the potential for irradiation damage in devices.

In 2018, XiaoLong Li [22] and colleagues from the Xinjiang Institute of Physics and Chemistry studied the mechanisms involved in using temperature switching as an accelerated test technique for enhanced low-dose-rate sensitivity (ELDRS). They used a specially designed gate-controlled lateral PNP transistor (GLPNP) to extract interface defects (N_it_) and oxide trapped charges (N_ot_). During irradiation with ^60^Co gamma rays, the electrical characteristics of the GLPNP transistor were measured in situ as a function of total dose. The results indicated that the generation of N_it_ in the oxide was the main cause of changes in the base current of the GLPNP. Based on the analysis of changes in N_it_ and N_ot_, the characteristics of accelerated proton release and inhibited proton loss, which occur with temperature switching, play a key role in determining the increase in N_it_ formation and the resulting degradation of base current as the dose accumulates. Additionally, the hydrogen dissociation mechanism, which leads to additional proton release and is related to the neutralization of N_ot_, further enhances the accumulation of N_it_. In the study, temperature-switching irradiation was used to conservatively estimate the ELDRS of GLPNP, providing a new method for ELDRS testing techniques.

In 2019, Zhao Jinyu [23] and colleagues from Harbin Institute of Technology studied the application of hydrogen immersion irradiation acceleration methods on 3DG111-type transistors and analyzed the radiation damage mechanism. The experimental results showed that the 3DG111-type transistor exhibited a significant enhanced low-dose-rate sensitivity (ELDRS) effect, with more severe degradation of electrical performance under low-dose-rate irradiation. The main reasons were the increase in oxide charge density and the shift of energy levels towards the mid-band. Transistors that underwent high-dose-rate irradiation after hydrogen immersion showed similar levels of electrical performance degradation as those under low-dose-rate irradiation. Therefore, hydrogen immersion treatment is an effective method for accelerating ELDRS effect assessment.

In 2020, Lei Dong [24] and colleagues from Harbin Institute of Technology studied the evolution of defects induced by ionizing radiation in gate-controlled lateral p-n-p (GLPNP) bipolar transistors at different temperatures. The experimental results showed that when the radiation dose was less than 25 krad, the change in the inverse of current gain (Δ(1/*β*)) of the transistor increased gradually with the rise in temperature. When the dose exceeded 25 krad, at 200 °C, Δ(1/*β*)) first increased and then decreased with the rise in temperature. Analysis using gate sweep (GS) technology and deep-level transient spectroscopy (DLTS) indicated that increasing temperature can reduce positive oxide trapped charges and increase the number of interface traps at temperatures below 200 °C, but the number of interface traps decreases at 200 °C.

The accumulation density of radiation-induced interface traps (N_it_) in GLPNP transistors at different temperatures and the variation with dose are shown in Figure 7. From the figure, it can be observed that the number of interface traps increases with the radiation dose at temperatures below 200 °C, and significantly anneals at 200 °C. As the temperature rises, positive oxide charges are more easily released into the traps, thereby increasing the number of H^+^ ions moving towards the interface, leading to an increase in the density of interface traps. Therefore, high temperatures can accelerate the formation of interface traps. When the temperature reaches 200 °C and the dose is not less than 25 krad, the interface traps significantly anneal, resulting in a decrease in their density.

In 2023, Hang Zhou [25] and colleagues from the Microsystem and Terahertz Research Center studied the effects of ionizing radiation on defect generation in GLPNP transistors at different temperatures and proposed a new model to explain the observed phenomena. The researchers conducted high-temperature radiation (ETI) experiments on the designed GLPNP transistors. Samples were irradiated at different temperatures (25 °C to 200 °C) to study the growth of interface traps at the SiO_2_/Si interface. In the high-temperature radiation experiments conducted over the temperature range of 25 °C to 200 °C, two optimal temperatures were identified. The new model suggests that these two optimal temperatures may respectively originate from the conversion of charge traps to interface traps and the yield of holes (or protons) from recombination. These findings provide clues to the evolution of defects in the gate oxide layer under ionizing radiation and also help understand the equivalence between high-temperature accelerated testing and real low-dose-rate radiation. The relationship between interface traps and irradiation temperature is shown in Figure 8. As shown in the figure, the response of interface traps to irradiation conditions has many unique characteristics. The density of interface traps first increases and then decreases with temperature, with two maxima occurring at 100 °C and 150 °C, respectively.

Key challenges include the complexity of defect dynamics, particularly the interactions between dangling bonds, oxygen vacancies, and hydrogen-related species. The role of hydrogen in defect formation and passivation, as well as the processes of hydrogen diffusion and release, are not fully understood. Additionally, determining appropriate acceleration factors and developing accurate quantitative models for defect behavior under stress conditions are difficult. Variability in material properties and device fabrication further complicates the generalization of findings across different technologies. Addressing these challenges requires advanced experimental techniques and computational modeling to achieve a comprehensive understanding of interface traps and their impact on device reliability.

### 2.3. The Mechanism of the Impact of Different Bias Conditions on Interface Traps

The impact of different bias conditions on interface traps in MOS devices is a critical area of study, as these traps significantly influence device performance and reliability. Interface traps, which form at the semiconductor–oxide interface, can alter key electrical properties such as threshold voltage, carrier mobility, and leakage current. Understanding how varying bias conditions—such as gate voltage, drain voltage, and substrate bias—affect the generation, distribution, and behavior of interface traps is essential for optimizing device design and ensuring long-term stability. This section explores the mechanisms by which different bias conditions influence interface traps and highlights the challenges in characterizing these effects.

#### 2.3.1. The Mechanism of the Impact of Different Bias Conditions on Interface Traps in MOS Devices

In 2021, Hugo Dewitte [26] and colleagues from the University of Toulouse in France studied the effects of extremely high total ionizing doses up to 450 Mrad(SiO_2_) on 180-nanometer MOSFETs in simulated applications, focusing on the impact of transistor size, design, type, bias, and annealing on radiation-induced degradation. The paper found that the gate bias of MOSFETs during radiation has a significant impact on the buildup of trapped charges in the oxide layer. The study also examined the effect of bias on leakage current, finding that the increase in leakage current under high-dose radiation is related to generation centers, which are produced by radiation-induced defects. The increase in leakage current is suppressed under reverse bias conditions, possibly due to the bias affecting the generation and distribution of defects, or due to local annealing effects. By studying the sub-threshold slope under different bias conditions, it was found that bias has a significant impact on the generation of interface states. The density of interface states is lower under normal bias conditions and higher under reverse bias conditions.

For the 10 μm and 10 μm STD p-MOS, the total ΔV_it_ and ΔV_ot_ displacements for three different biases are shown in Figure 9. The results show that under normal bias, the threshold voltage offset is mainly caused by oxide traps (ΔV_ot_), while under reverse bias, the effect of interface traps (ΔV_it_) is more significant. Under ground bias, the offsets of both interface traps and oxide traps are relatively low, which may be due to the lower charge generation rate, reducing the formation of defects. These data emphasize the importance of bias conditions on the performance of MOSFETs in radiative environments.

The major challenge is the dynamic nature of interface traps under varying bias conditions, which complicates the identification of specific defect types and their evolution. Additionally, the interplay between electric fields, carrier injection, and defect reactions under bias is not fully understood, making it difficult to predict trap behavior accurately. Another issue is the lack of comprehensive models that can account for the combined effects of multiple bias conditions on interface trap formation and dynamics. Addressing these challenges requires advanced experimental techniques and computational approaches to provide deeper insights into the mechanisms driving bias-dependent interface trap behavior.

#### 2.3.2. The Mechanism of the Impact of Different Bias Conditions on Interface Traps in Linear Bipolar Devices

In 2016, Jinxin Zhang [27] and colleagues from Xi’an Jiaotong University conducted research on the impact of ^60^Coγ-rays on the total ionizing dose (TID) effect in silicon–germanium heterojunction bipolar transistors (SiGe HBTs) under different bias conditions. The experiments showed that the TID effect is caused by radiation-induced defects in the oxide layer, which include positive oxide traps (N_ot_) and interface states (N_it_). Different bias conditions lead to different distributions of N_ot_ and N_it_ in the oxide layer, resulting in varying degrees of damage. As the total dose accumulates, the contribution of N_it_ to radiation damage becomes more significant.

In 2021, Zhuoqi Li [28] and colleagues from the Department of Nuclear Science and Technology at Xi’an Jiaotong University studied the degradation characteristics of SiGe HBTs after multiple-pulse neutron-gamma irradiation under different bias modes (forward, cutoff, and saturation). The test samples were silicon–germanium NPN RF transistors (BFU730F) manufactured using 0.25-micron SiGe:C technology. The irradiation experiments were conducted at the Xi’an Pulse Neutron Reactor, with three common operating bias conditions set: forward bias, cutoff bias, and saturation bias. As shown in the Figure 10, the degradation degree of the test samples varied under different bias conditions, with the samples under cutoff bias showing the most severe degradation, followed by those under saturation bias and forward bias. The differences in degradation under different bias conditions were mainly related to the electron occupation of relevant traps, which corresponded to the free charge injection annealing effect.

Figure 10 shows the change in the reciprocal of current gain Δ(1/*β*) of silicon–germanium heterojunction bipolar transistors (SiGe HBTs) as a function of neutron fluence under different bias conditions, with a fixed base-emitter voltage (V_BE_) of 0.7 volts. SiGe HBTs under forward bias exhibit the smallest change in the reciprocal of current gain, while those under cutoff bias show the largest change. SiGe HBTs under saturation bias have a change in the reciprocal of current gain that lies between the two. Forward bias may alleviate irradiation-induced performance degradation by promoting the annealing of defects. In contrast, cutoff bias may lead to more severe performance degradation due to the lack of injected current to facilitate the reconfiguration of defects, resulting in more defects being trapped and stabilized.

In 2022, Peng Chao [29] and colleagues from the Fifth Research Institute of Electronics and Information Industry conducted total dose irradiation experiments to study the radiation damage effects of domestic NPN bipolar transistors under different biases and analyzed the performance degradation caused by radiation-induced defects. Two structurally identical silicon epitaxial planar NPN transistors were selected as test samples, one of which had undergone radiation-hardening treatment. A ^60^Coγ-ray was used as the radiation source with a dose rate of 0.1 rad(Si)/s and a total dose of 50 krad(Si). During the experiment, the devices were irradiated under two different bias conditions. The experimental results showed that irradiation led to an increase in the base current and a decrease in the current gain of the bipolar transistors. The NPN transistors showed more severe degradation under reverse bias than under zero bias. Radiation-induced defects mainly led to an increase in recombination current, thereby causing an increase in base current. Through deep-level transient spectroscopy (DLTS) testing, it was found that radiation leads to an increase in defect density and a change in defect energy levels.

## 3. Overview on the Annealing Mechanism of Interface Traps

The annealing mechanism of interface traps is a critical aspect of semiconductor device reliability, as it directly influences the recovery and stability of device performance after exposure to stress conditions such as radiation or high-temperature operation. Annealing, which involves the thermal treatment of devices, can lead to the passivation or removal of interface traps, thereby restoring electrical properties such as threshold voltage and carrier mobility. Understanding the underlying mechanisms of interface trap annealing, including the role of temperature, time, and environmental factors, is essential for developing effective strategies to enhance device longevity and performance. This section provides an overview of the current understanding of interface trap annealing mechanisms and highlights the key challenges in this area.

### 3.1. Research on the Annealing Mechanism of Interface Traps

#### 3.1.1. Research on the Annealing Mechanism of Interface Traps in MOS Devices

In 2011, D. R. Hughart [30] and others from Vanderbilt University described a hydrogen dimerization mechanism, which helps to explain the accumulation and annealing of interface traps during high-temperature irradiation. The most important factor that determines whether interface trap formation or passivation will occur is the concentration of protons near the interface. Hydrogenated oxygen vacancies react with protons to generate molecular hydrogen, facilitating the hydrogen dimerization. At high temperatures, this mechanism consumes a large number of protons accumulated near the interface, limiting the supply of protons available for interface trap formation. Since the concentration of protons determines the reaction at a given temperature, by examining how factors such as dose rate and total dose affect the proton concentration near the interface, their impact on the accumulation of interface traps can be explained.

In 2018, Xiaojing Li [31] et al. from the Institute of Microelectronics, Chinese Academy of Sciences emphasized the effectiveness and sensitivity of the direct current–current–voltage (DCIV) method in monitoring the interface trap density in deep submicron MOSFETs. Moreover, through experimental and theoretical analyses, they revealed the generation and variation mechanisms of interface traps under F-N stress conditions. The interface trap density and distribution at the Si/SiO_2_ interface in partially depleted SOI MOSFETs were measured before and after F-N tunneling stress by using the direct current–current–voltage (DCIV) method. An effective method was proposed to evaluate the interface state of SOI devices subjected to electrical stress or other damages. By combining the DCIV measurement results with theoretical calculations, the equivalent density and energy level of the interface traps were obtained using the least squares method. It was found that the interface trap density increased with the application of F-N stress due to the generation of Si dangling bonds and the trapped charges on the Si/SiO_2_ interface. As the stress time increased, the equivalent energy level of the interface traps approached the mid-gap. These findings are of great significance for understanding and improving the reliability of SOI MOSFETs under electrical stress.

In 2019, Stefano Bonaldo [32] from the Department of Information Engineering, University of Padua, Italy investigated the charge accumulation and spatial distribution of interface traps in 65 nm pMOSFETs under ultrahigh-dose radiation as well as their degradation behaviors under different bias conditions. A commercial 65 nm CMOS technology was used in the experiment, and the samples were irradiated by X-rays and then annealed at high temperature under different bias conditions. In the experiment, static DC and charge-pumping measurements were carried out on the pMOSFETs. The experimental results and 3D TCAD simulations demonstrated that pMOSFETs are highly sensitive to the radiation-induced short-channel effect, which is related to the defect accumulation in the spacer dielectrics. The experiment also found that the charge accumulation in the spacer is independent of the applied source-to-drain electric field, but the generation and annealing of interface traps strongly depend on the applied drain bias and channel length.

In 2024, Yu Song [33] et al. from Neijiang Normal University proposed a hydrogen anion passivation mechanism for interface trap annealing (ITA) in forward-biased metal-oxide semiconductor (MOS) devices. Experiments showed that the protons released due to cracking on the oxide traps could migrate into silicon at a relatively high temperature, thereby passivating the Pb centers at the interface by capturing the two electrons provided by the forward bias. This new mechanism explains the elimination temperature of approximately 100 °C in the experiment, which is much lower than the temperature exceeding 220 °C predicted by the traditional passivation mechanism. Based on the coupled conversion kinetics between oxide and interface traps and the concept of hierarchical confinement kinetics in glassy materials, an analytical model of ITA was derived. These models can generally describe the transition from the generation to the elimination of Pb and predict the enhanced elimination observed when the annealing temperature is increased. The defect annealing model can be applied to analyze the non-monotonic temperature dependence of the threshold voltage shift annealing in nMOS devices. This finding provides a theoretical basis for eliminating interface traps at low temperatures and theoretical support for setting the HT stage of accelerated aging and damage assessment of MOS devices. A schematic diagram of the H^+^-mediated transformation and H^−^-induced passivation-related processes is shown in Figure 11.

One of the primary challenges in studying the annealing mechanism of interface traps is the complexity of the processes involved, which can include defect diffusion, recombination, and passivation by species such as hydrogen. The precise pathways and kinetics of these processes are not fully understood, making it difficult to predict the effectiveness of annealing under different conditions. Additionally, the interaction between interface traps and bulk defects during annealing adds another layer of complexity. Another challenge is the variability in annealing outcomes due to differences in material properties and device fabrication processes. Addressing these challenges requires a combination of advanced experimental techniques and theoretical modeling to gain a comprehensive understanding of the annealing mechanisms and their impact on device performance.

#### 3.1.2. Research on the Annealing Mechanism of Interface Traps in Linear Bipolar Devices

In 2019, Jianqun Yang [34] et al. from Harbin Institute of Technology studied the annealing effect of displacement damage in NPN transistors due to ionization damage. The electrical degradation and radiation defects of F2N2219 NPN transistors caused by independent and sequential irradiations of 40 MeV silicon ions and cobalt-60 gamma rays were characterized. The experimental results showed that the current gain degradation of F2N2219 bipolar junction transistors (BJTs) under the sum of independent cobalt-60 gamma radiation and heavy ion irradiation was higher than that under sequential irradiation, indicating a significant synergistic effect. Based on the deep-level transient spectroscopy (DLTS) analysis, it was shown that the displacement defects caused by 40 MeV silicon ions were annealed due to the oxide charges induced by cobalt-60 gamma-ray irradiation. Compared with VO and V_2_(=/-) centers, the V_2_(-/O) center was more susceptible to oxide charges, resulting in significant annealing. In addition, the larger the fluence of independent 40 MeV silicon ions, the more obvious the annealing effect of displacement defects. Figure 12 shows the relationship between the excess base current and the emitter-base voltage of bipolar junction transistors (BJTs) irradiated sequentially by 40 MeV silicon ions and cobalt-60 gamma rays. It can be seen that as the dose of cobalt-60 gamma rays increased, the excess base currents of F2N2219 transistors irradiated by 40 MeV silicon ions with all different fluences increased. Moreover, for the lower fluence of 40 MeV silicon ions, when the dose of cobalt-60 gamma rays was low, the excess base current of the transistors irradiated sequentially was less than that of the transistors irradiated by independent 40 MeV silicon ions. The above results indicate that the ionization damage to the transistors caused by subsequent cobalt-60 gamma rays has an annealing effect on the displacement damage caused by 40 MeV silicon ions, and the larger the displacement damage, the more obvious this annealing effect.

### 3.2. Variable-Temperature Annealing Experiments After Irradiation

Variable-temperature annealing experiments after irradiation are crucial for understanding the recovery mechanisms of radiation-induced defects in semiconductor devices. By systematically varying annealing temperatures, researchers can study the kinetics of defect passivation and the restoration of device performance. These experiments provide insights into how thermal energy influences the stability and behavior of interface traps, offering a pathway to improve radiation hardness and reliability.

#### 3.2.1. Variable-Temperature Annealing Experiment After Irradiation in MOS Devices

A key challenge lies in the complexity of defect dynamics during annealing, as different defect types may exhibit varying responses to temperature changes. Additionally, the interaction between defect recovery processes and material properties complicates the prediction of annealing outcomes. Understanding the optimal annealing conditions and their impact on different device architectures remains a significant hurdle, requiring advanced experimental and modeling approaches.

In 2005, X. J. Zhou et al. from Vanderbilt University [35] investigated the combined effects of irradiation and bias temperature stress (BTS) on MOS capacitors with HfO2 dielectrics. It was found that irradiation significantly enhances BTS-induced degradation, with the extent of enhancement varying depending on the irradiation bias. Specifically, zero-bias or positive-bias irradiation followed by negative BTS (NBTS) results in much worse degradation compared to either irradiation or NBTS alone. This phenomenon is primarily attributed to the formation of dipoles during irradiation and the electrostatic repulsion of electrons from the oxide during NBTS. In integrated circuit applications, the worst-case scenario is expected for pMOS transistors irradiated in their “off” states and annealed in their “on” states. For comparison, the study also examines Al_2_O_3_-based MOS capacitors with Al gates and SiO_2_-based MOS capacitors with NiSi gates. The Al_2_O_3_-based devices show somewhat less sensitivity to the combined effects of irradiation and BTS, while the thermal oxides with NiSi gates exhibit significantly less sensitivity. These findings highlight the critical role of material properties and device architecture in determining the response to irradiation and BTS, providing valuable insights for the design of radiation-hardened semiconductor devices.

#### 3.2.2. Variable-Temperature Annealing Experiments After Irradiation in Linear Bipolar Devices

In 2004, Marty R. Shaneyfelt et al. from Sandia National Laboratories [36] investigated the annealing behavior of LM139 comparators fabricated using National Semiconductor Corporation (NSC) enhanced low-dose-rate-sensitive (ELDRS) linear bipolar technology after irradiation, especially its impact on total dose degradation. The data showed that a large portion of the radiation-induced increase in input bias current was recovered after annealing at 100 °C. The recovery of the input bias current was related to significant interface trap annealing at 100 °C. This was qualitatively consistent with previous interface trap annealing data and the latest models of interface trap annealing related to hydrogen movement at the silicon/silicon dioxide interface. These data helped to explain why high-dose-rate irradiation at high temperatures could underestimate low-dose-rate degradation in some cases. In addition, the data confirmed that high-dose-rate irradiation followed by high-temperature annealing did not simulate the mechanism that leads to enhanced degradation at low dose rates in devices with ELDRS. Figure 13 shows the results of annealing at 100 °C after low-dose-rate and high-dose-rate irradiation and room temperature annealing. The annealing behavior indicated that there was only a small change in the build-up or annealing of interface traps in the devices irradiated at 211 rad(SiO_2_)/s. However, the 100 °C annealing had a significant effect on ΔI_B+_ of the devices irradiated at 0.01 rad(SiO_2_).

In 2006, X. J. Chen et al. from Arizona State University [37] studied the nature of radiation-induced switching state establishment in ELDRS bipolar devices by using low-dose-rate radiation as a function of bias voltage and post-radiation annealing temperature. It was observed that both interface traps and border traps contributed significantly to the radiation-induced damage in linear bipolar devices. For devices irradiated at 0 V bias, the switching state was mainly manifested as interface traps. For devices under −50 V irradiation bias, a combination of interface traps and border traps was found. Radiation annealing indicated that the interface traps in the devices irradiated at 0 V were removed at temperatures below 100 °C. In the devices irradiated at −50 V, the radiation annealing of border traps and the increase in interface trap density were observed. These results confirmed the importance of interface traps and border traps to the radiation response of linear bipolar transistors, and demonstrated that there might be significant differences in the annealing responses of these defects. Figure 14 shows the corresponding annealing results of the −50 V irradiated bias devices. Judging from the amplitude of the current peak, the data suggested a small increase in the density of the switching state after annealing at different temperatures. The rightward shift of the current curve indicated that the charges captured in the oxide were annealed.

In 2015, Chaoming Liu [38] et al. from Harbin Institute of Technology used electrical characteristics and deep-level transient spectroscopy (DLTS) to measure the radiation defects caused by ionization and displacement damage during the annealing process. In 3CG110 PNP bipolar junction transistors (BJTs), a nonlinear relationship between proton fluence and radiation response was clearly observed. The DLTS analysis technique and the annealing response of BJTs can provide important information about the nature of ionization- and displacement-induced defects and quantitatively measure them, especially for BJTs with combined radiation damage caused by protons. Based on the DLTS measurement results and the results of current gain annealing, the evolution of ionization and displacement defects during irradiation and annealing was revealed, and the relationship between the defects and current gain annealing was investigated. Figure 15 shows the variation of the current gain of 3CG110 PNP transistors with the annealing temperature. According to the results in the figure, when the annealing temperature is lower than 600 K, the current gain recovers slightly. In contrast, when the annealing temperature > 600 K, the current gain increases rapidly with the increase of temperature. When the annealing temperature reaches 700 K, the current gain almost returns to the state before the transistor was irradiated.

In 2018, Xiaolong Li and Wu Lu [39] et al. from Xinjiang Technical Institute of Physics & Chemistry, Chinese Academy of Sciences developed a temperature-switching irradiation (TSI) sequence based on the first-principles understanding of interface trap formation and annealing. The authors described in detail the dynamics of interface trap formation and annealing during high-temperature irradiation. The TSI method has been proven to be a practical and conservative testing method for detecting ELDRS in linear bipolar devices and integrated circuits. The basic principle of TSI is based on the kinetic understanding of interface trap formation and annealing. At high temperatures and low doses, the interface trap density is relatively low, and high-temperature irradiation mainly accelerates the transport of protons to the Si/SiO_2_ interface. A schematic diagram of the TSI method for characterizing ELDRS in linear bipolar technology is shown in Figure 16. It includes a four-step irradiation procedure: (1) the sample is irradiated at a dose rate of 120 °C (Si)/s (20 krad (Si)); (2) it is then irradiated at 100 °C to 40 krad (Si); (3) it reaches 80 krad (Si) at 65 °C at the same rate; and (4) finally, it reaches 100 krad (Si) at 50 °C at the same rate.

In 2022, Delgermaa Nergui [41] et al. from Georgia Institute of Technology investigated the total ionization dose response of the third-generation SiGe HBT (Silicon Germanium Heterojunction Bipolar Transistor) at relatively high temperatures (80 °C and 130 °C) and compared it with the response at 30 °C. These devices exhibited less radiation damage at higher temperatures. The degradation mechanisms were studied through annealing experiments and TCAD simulations. The reduction in forward-mode degradation at higher temperatures was mainly due to the increased annealing of interface traps resulting from the diffusion and reaction of H_2_ within the oxide/nitride EB (emitter-base) spacer. The degradation was moderate at all measured temperatures, making these devices suitable for radiation environments where higher temperatures are expected. Figure 17 shows the estimated N_it_ due to radiation dose and after high-temperature baking at 100 °C and 130 °C. Although the annealing rate at 130 °C is faster than that at 100 °C, the overall trends at the two temperatures are similar. When the devices were irradiated and annealed at high temperatures with all pins grounded, there was clear evidence of significant annealing of interface traps, which is consistent with many previous studies [42].

In 2023, Rigen Mo [43] et al. from the China Academy of Space Technology took GLPNP transistors as the research object to study the annealing process of bipolar transistors after total dose irradiation. After total dose irradiation, room temperature and high temperature annealing tests were carried out. The Gummel curves, GS curves, and SS curves of bipolar transistors were obtained through I/V scanning, gate scanning, and subthreshold scanning. Through different microscopic defect characterization methods, the evolution behaviors of radiation-induced oxide trap charges and interface state damage defects during the annealing process of bipolar transistors at different temperatures were quantitatively analyzed. This research is of great significance for a better understanding of the recovery mechanism of the electrical performance of bipolar transistors during annealing after total dose irradiation, and can provide references for the design, application, and radiation hardening of transistors.

## 4. Overview of the Simulation Research on Interface Traps

In the past few decades, computational simulations have become an important tool for studying interface defects. From first-principles calculations to multiscale simulations, various computational methods have been used to predict and understand the physical and chemical properties of interface defects. With the improvement of computing power and the development of algorithms, these simulation studies can provide increasingly accurate information about defect structures and electronic properties. In addition, with the development of machine learning technology, data-driven methods have also begun to be applied to the study of interface defects, providing new perspectives and tools for traditional computational simulations. This section reviews the latest progress in computational simulations of interface defects, including first-principles-based calculations, multiscale simulation methods, and the application of machine learning technology in the study of interface defects, and explores how these methods help us study the formation mechanisms, transformation processes, and impacts on device performance of interface traps.

The simulation research of interface defect calculation has witnessed rapid development. From the early simple models to the current high-precision calculations based on first principles, the continuous progress of simulation techniques has led to a deeper understanding of defect properties. Researchers have used methods such as density functional theory (DFT) and molecular dynamics (MD) to discuss in detail the formation energy and activation energy of interface defects and their impacts on material properties. The simulation research of interface defects is developing towards high-precision, multiscale, and comprehensive applications. Future research will further deepen, covering a wider range of material systems and complex environments.

### 4.1. Analytical Models and Simulation Methods

Analytical models and simulation methods have been widely employed to study the radiation response of semiconductor devices, providing valuable insights into the mechanisms of radiation-induced damage and its impact on device performance. These approaches, ranging from empirical models to advanced numerical simulations, enable the prediction of key parameters such as threshold voltage shifts, leakage currents, and degradation rates under various radiation conditions. By combining experimental data with theoretical frameworks, researchers have developed foundational models to describe the general behavior of devices exposed to ionizing radiation. However, despite these advancements, significant gaps remain in understanding the variability in radiation damage across different devices and the underlying factors influencing their sensitivity to dose rates.

In 2015, Chenhui Wang [44] et al. from Northwest Institute of Nuclear Technology used TCAD(2013) to conduct a numerical simulation of the ionization/displacement synergy effect of mixed neutron and gamma-ray radiation on six different types of lateral PNP bipolar transistors. Lateral PNP bipolar transistors were selected as the simulation devices, and the neutron displacement effect and total ionization dose effect were simulated in TCAD. The article analyzed the physical mechanism of the ionization/displacement synergy effect and pointed out that the positive charges induced by gamma rays in the oxide layer and Si/SiO_2_ interface traps can enhance the carrier recombination process in the bulk defects caused by neutron irradiation, which is the main reason for the ionization/displacement synergy effect of lateral PNP bipolar transistors. Figure 18 shows the change in the reciprocal of the common-emitter current gain caused by neutron fluence in lateral PNP bipolar transistors under mixed neutron and gamma radiation irradiation. The simulation results in the figure indicate that the ionization/displacement synergy effect is not a simple addition of the total ionization dose effect and the displacement effect. The total ionization dose effect can act jointly with neutron displacement damage, resulting in greater gain degradation.

In 2018, B. S. Tolleson [45] et al. from Arizona State University proposed an analytical model to describe the excess base current in lateral PNP bipolar junction transistors. The paper presented a new model that more accurately predicts the excess base current in LPNP BJTs by calculating the influence of charged interface traps on the base surface potential and surface carrier concentration.

In 2019, Ruibin Li [46] and colleagues from Xi’an Jiaotong University conducted experiments to explore the synergistic effects of total ionizing dose (TID) and Analog Transient Radiation Effects in Electronics (ATREE) in vertical NPN (VNPN) bipolar transistors. They used computer-aided design (TCAD) simulations to analyze the impact of TID-induced oxide layer positive charges and Si/SiO_2_ interface traps on the primary and secondary photocurrents of the transistor under transient ionizing radiation conditions. The 2N2222A transistor was used for the radiation experiments, which were divided into four groups, with three groups receiving different doses of TID radiation and one group remaining unirradiated. After TID radiation, the collector current and base current of the transistor were immediately measured to conduct transient ionizing radiation experiments and obtain the transistor’s response to ATREE. The results showed that a TID-ATREE synergistic effect was observed in VNPN bipolar transistors, where an increase in TID led to faster decay of the secondary photocurrent and shorter transient duration under pulsed X-ray radiation. The TID-induced Si/SiO_2_ interface traps were the main factor in the synergistic effect, while the positive charges in the oxide layer had a smaller impact.

In 2020, Li Lei [47] et al. from the Institute of Nuclear Physics and Chemistry, China Academy of Engineering Physics proposed an analytical model to describe the defect generation and the change in the extra base current (ΔI_B_) of PNP BJTs under the influence of total ionization dose, dose rate, and hydrogen concentration. The model consists of three key parts: the dynamics of oxide layer charges (Node I), interface traps (Node II), and extra base current (Node III), which can fit the experimental data well, indicating that the generation of oxide layer charges and interface traps is related to the total ionization dose, dose rate, and hydrogen concentration. The research results show that the shallow traps are crucial for the generation of interface traps related to the space charge mechanism. The proposed model is applicable to devices with good-quality SiO_2_ and can provide predictions of the extra base current for PNP BJTs at low dose rates.

The interface trap density responses of GLPNP to dose rate and hydrogen concentration after irradiation damage were predicted, as shown in Figure 19. Figure 19 compares the experimental results of interface trap density with the fitting of the theoretical model in an air environment (Air), 1% hydrogen environment (1% H_2_), and 100% hydrogen environment (100% H_2_). The results show that under all test conditions, the interface trap density decreases with the increase in dose rate, and in a higher hydrogen concentration, the interface trap density increases significantly. This indicates that the presence of hydrogen significantly enhances the interface trap generation of the transistor in the radiation environment, which may affect the long-term stability and performance of the device.

In 2021, Yu Song [48] et al. from the Research Center of Microsystems and Terahertz derived a new analytical model to describe the relationship between defect concentration and irradiation dose and tested it through gamma-ray irradiation experiments. It was proposed that the generation of E’ was slowed down due to the dispersive diffusion of holes induced in disordered silica, while the transformation of P_b_ became reversible due to the defect reaction with enhanced recombination under irradiation. The research shows that the analytical model derived based on these new understandings can consistently explain the dependence of defect concentration on dose and dose rate in a wide range.

Current research has primarily focused on establishing basic models to describe radiation damage, but these models often fail to account for the observed differences in radiation damage magnitude and sensitive dose rates among various devices. The reasons for such variability, including material properties, device geometry, and fabrication processes, are not well understood. Furthermore, the formation processes of interface defects, which play a critical role in determining device radiation sensitivity, remain poorly characterized. Key factors such as the influence of interfacial stress, impurity concentrations, and defect interactions on defect formation and evolution are yet to be accurately explained. This lack of understanding limits the ability to predict and mitigate radiation-induced degradation in specific device types, particularly under low-dose-rate conditions where defect dynamics are highly complex. Addressing these limitations requires a more nuanced approach that integrates detailed material characterization, advanced simulation techniques, and experimental validation to uncover the fundamental mechanisms driving device-specific radiation responses.

### 4.2. Deep Learning and Optimization Algorithms

In the prediction of irradiation damage, traditional simulation and experimental methods usually consume a great deal of time and resources. Machine learning can extract key features and patterns from historical experimental data by training models, thus improving the accuracy and efficiency of prediction. Machine learning models can replace time-consuming TCAD calculations to achieve acceleration and predict the damage of transistors under different dose rates. Moreover, machine learning methods can handle higher dimensional and nonlinear data features, making them more promising in dealing with complex interface traps research. Although some studies have utilized machine learning to analyze irradiation effects, a systematic study of the types of interface traps and their formation processes remains an area worthy of in-depth exploration.

In 2022, B. Dean [49] et al. from Vanderbilt University used a convolutional neural network (CNN) in deep learning to analyze the total ionizing dose (TID) effects of Commercial Off-The-Shelf (COTS) electronic products. The Texas Instruments’ MSP430FR6989 Microcontroller Unit (MCU) was used in the study, and it was irradiated with 10 keV X rays under different low-power modes (LPMs). The CNN model was trained by measuring the internal noise signatures generated by the clock module after irradiation to perform classification and regression analysis on the MCU and predict the TID.

In 2022, BaiChuan Wang [50] et al. from Tsinghua University proposed a machine learning based scientific discovery method, using an artificial neural network (ANN) model to analyze the experimental data of the impact of TID on BJTs. A three-layer ANN model was used, and trained with the dropout algorithm and the Adam optimizer. The trained ANN model outperformed traditional multiple linear regression in capturing nonlinear correlations and predicting data. The model indicated that the TID hardness of BJTs increases with the increase in the base current I_B0_. Radiation experiments verified that the emitter perimeter to area ratio might be one of the reasons for this phenomenon. Figure 20 shows the schematic diagram of the proposed artificial neural network model. The dataset used contains 565 radiation response data points from 10 articles, involving two types of BJTs, NPN and LPNP. The input parameters of the dataset include bias, layout, dose, dose rate, passivation layer, hydrogen content, and other parameters, and the output is a single neuron corresponding to the logarithm of the base current change. The schematic of the proposed ANN model is shown in Figure 20.

In 2022, Li Lei [51] et al. from the China Academy of Engineering Physics proposed an artificial neural network (ANN) model based on deep learning (DL) technology to calculate the density of ionized defects. This model can be trained using data prepared by combining physical models with numerical calculations. A dataset of approximately 6.0 × 10^5^ data points was prepared through physical models and numerical calculations for training and testing the ANN model. The model inputs include dose, dose rate, and hydrogen concentration, and the outputs are the densities of oxide charges and interface states. The paper studied the structure of the ANN model, data preprocessing methods, neurons and outputs, training methods, and case studies. By comparing different network configurations, an optimized configuration was selected to calculate the accumulation of ionized defects in BJTs. The paper also explored the influence of the number of model parameters, the size of the training dataset, and the activation function on the model performance. The optimized ANN model consists of four hidden layers and sixteen neurons per hidden layer activated by the exponential linear unit (ELU). This model can fit the data from the physical model well, including the effects of dose, dose rate, and hydrogen concentration on ionized defects. The structure and data flow of the artificial neural network used in the paper are shown in Figure 21. It is a feed-forward network with n layers, consisting of one input layer, n − 2 hidden layers, and one output layer.

In 2024, Chen Chong [52] et al. from Xidian University proposed an algorithmic prediction of the single particle radiation effect of a novel tunnel field effect transistor (TFET). The authors constructed a deep learning algorithm network model to predict the key characteristic parameters of single particle transients in TFET devices. Computer-aided design technology was used to study the impact of single particle effects on the novel stacked source channel gate TFET devices. The deep learning network prediction model showed a relative error percentage of the predicted values of less than 1%, indicating excellent prediction results. Compared with support vector machines, decision trees, K-nearest neighbor algorithms, ridge regression, and linear regression, the deep learning algorithm had the smallest average relative error percentage.

The simulation research on interface traps is in a stage of rapid development. The progress of computational methods and the application of machine learning techniques have provided new possibilities for understanding and managing interface defects. The advancements in the simulation of interface defect calculations enable researchers to gain in-depth insights into the formation and evolution mechanisms of defects at the micro level, thus providing a solid theoretical basis. The machine learning-based research on interface defects further broadens the research methods and offers powerful tools for the rapid identification, prediction, and optimization of defect characteristics. In the future, with the combination of theory and experiment and the further improvement of computing power, this field is expected to achieve a higher level of innovation, providing a more solid foundation for the development of new materials and devices.

### 4.3. First-Principles Calculations

The study of radiation-induced defects using first-principles calculations has gained significant attention in recent years due to its critical role in understanding the degradation mechanisms of semiconductor devices under radiation exposure. First-principles methods, such as density functional theory (DFT), provide a powerful tool for investigating the atomic-scale properties of defects, including their formation energies, charge states, and migration pathways. These insights are essential for predicting the behavior of radiation-induced defects and their impact on device performance. However, despite the progress made in characterizing defect structures and energetics, significant challenges remain in accurately quantifying the kinetic parameters associated with defect reactions, such as reaction rates and activation energies. This limitation hinders a comprehensive understanding of defect dynamics under operational conditions, particularly in complex environments involving multiple defect types and interactions.

In 1999, Sokrates T. Pantelides [53] et al. from Vanderbilt University conducted in-depth research on the atomic scale dynamics during the oxidation of silicon, as well as the formation and properties of defects at the silicon and silicon dioxide interface through first-principles calculations. The research team identified the mechanism of silicon interstitial atom emission. This mechanism can eliminate electrically active defects without introducing dangling bonds, thus providing an explanation for the low defect density at the silicon and silicon dioxide interface. In addition, the paper proposed a new class of electrically active interface traps, which are structurally similar to thermal donors in silicon. These findings are of great significance for understanding the oxidation behavior of silicon in microelectronic applications and interface traps.

In 2001, S. N. Rashkeev [54] et al. studied the first-principles calculations of interface trap formation in MOS structures. It was determined that H^+^ is the only stable charge state of hydrogen at the interface, and H^+^ reacts directly with Si-H. The product of this reaction is an H_2_ molecule and a positively charged dangling bond center (D^+^), formed through the reaction SiH + H^+^ → D^+^ + H_2_. Density functional theory calculations indicate that a migrating proton cannot capture an electron when approaching the interface [55]. The reaction between H+ and an interfacial Si-H bond is shown in Figure 22.

In 2011, Nicole L. Rowsey [56] et al. from the University of Florida developed a physics-based TCAD model to investigate the enhanced low-dose-rate sensitivity in linear bipolar devices. Over a broad range of dose rates and hydrogen concentrations, this model was in quantitative agreement with the measurement data. Through the analysis of the degradation effects of individual defect types, which was guided by first-principles calculations, insights into the mechanisms underlying the enhanced low-dose-rate effects in different hydrogen environments were provided. The impacts of the initial defect concentration and location, as well as the energetics of defect related reactions, were studied, and conclusions were reached regarding the roles of molecular hydrogen and hydrogenated defects in the radiation response of these devices. The simulation results in Figure 23 demonstrate a quantitative match with the data from Chen et al. in terms of the relationship between the interface trap density and the ambient hydrogen concentration.

In 2022, Haoran Zhu [58] et al. from the Microsystem and Terahertz Research Center, China Academy of Engineering Physics calculated the temperature-dependent carrier capture coefficients of interface defects P_b0_ and P_b1_ at the amorphous SiO_2_/Si(100) interface by combining first-principles and one-dimensional static coupling theoretical methods. It was found that after P_b0_ and P_b1_ defects capture carriers to form charged defects, their geometries undergo significant deformation, especially for silicon atoms containing dangling bonds. The hole capture coefficients of neutral P_b0_ and P_b1_ defects are larger than other capture coefficients, indicating that these defects are more likely to form positive charge centers. Meanwhile, the calculated results of the nonradiative recombination coefficients of these defects show that both P_b0_ and P_b1_ defects are the main nonradiative recombination centers at the amorphous SiO_2_/Si(100) interface.

In 2023, Zenghui Yang [59] et al. from the Research Center of Microsystems and Terahertz, China Academy of Engineering Physics studied the effects of γ-ray irradiation on defects in silicon transistors through a multiscale simulation method, combining Monte Carlo simulation and excited state first-principles calculations. By developing a multiscale method to simulate γ-ray irradiation, a defect-based model of the irradiation synergy effect (ISE) was examined, in which the γ-ray-induced electron excitations were explicitly treated in the excited state first-principles calculations. It was found that the calculation results were consistent with the experimental results, and the γ-ray-induced excitation effects were significantly different from the effects of defect charge states and temperature. Finally, a diffusion-based qualitative explanation was proposed to explain the mechanisms of positive/negative ISE in NPN/PNP BJTs.

Currently, the quantitative calculation of kinetic parameters related to defect reaction dynamics remains a significant challenge. While first-principles methods have been successful in elucidating the static properties of defects, accurately modeling the time-dependent evolution of defects, including their formation, migration, and annihilation processes, requires overcoming substantial computational and theoretical hurdles. These include the need for more advanced techniques to simulate rare events, account for temperature effects, and incorporate the influence of external factors such as electric fields and stress. As a result, there is a critical gap in our ability to predict the long-term behavior of radiation-induced defects and their cumulative impact on transistor performance, particularly under low-dose-rate conditions where defect kinetics play a dominant role. Addressing these limitations is essential for advancing the reliability and radiation hardness of semiconductor devices in harsh environments.

## 5. Challenges and Future Outlook

From the literature review, it can be seen that the properties of interface defects and their formation processes have been studied in the industry, and the influence of interface defect concentration on the electrical parameters of transistors has also been investigated. However, the impact of the types of interface defects and their formation processes on the electrical parameters of transistors remains unclear. The main type of interface state is a silicon dangling bond, and the interface is quite complex, with properties that are not those of a single defect. Therefore, it is necessary to clarify their types and formation processes, focusing on their microscopic changes.

The microscopic structure of radiation-induced interface trap charges has been identified as dangling silicon bonds at the Si/SiO_2_ interface. Gamma radiation-induced protons contribute to the formation of interface defects, and the extent of device performance degradation is influenced by the concentration of these defects. Industry-proposed models have revealed the mechanism behind the enhanced low-dose-rate sensitivity (ELDRS) effect, highlighting the competition between proton release reactions involving hydrogen-related defects or hydrogen molecules and the neutralization reactions of defect-trapped electrons as the key factor driving ELDRS. Based on first-principles defect reaction kinetics, the microscopic structure, energy levels, and reaction processes of interface defects in bipolar devices can be calculated, enabling a shift from qualitative analysis to quantitative research and even scientific prediction in ELDRS studies. For linear bipolar devices operating in radiation environments, the most effective approach is to conduct radiation hardness assurance tests at low practical dose rates (typically 10 mrad(SiO_2_)/s) under varying temperatures and doses.

The properties of interface defects are not uniform, and both their nature and formation processes remain poorly understood in terms of their impact on the electrical parameters of transistors. While the previous literature has primarily focused on dose rates above 10 mrad(SiO_2_)/s, damage is observed to further increase at dose rates below 1 mrad(SiO_2_)/s, yet the evolution of dominant radiation-induced defects at these lower dose rates lacks a precise explanation. Quantitative determination of the kinetic parameters associated with defect reactions remains challenging, and the magnitude of radiation damage and sensitive dose rates vary across different devices. Although basic models have been established, the reasons for device-specific differences and the factors influencing interface defect formation are not well understood. Furthermore, the mechanisms of radiation effects on transistors with different structures (e.g., LPNP, SNPN, and VPNP), as well as the impact of variations in size, shape, doping, and configuration (e.g., lateral, vertical, and composite) within the same transistor type, remain inadequately explained.

This thesis aims to conduct an in-depth study of the formation and annealing processes of interface defects in microelectronic devices under irradiation environments, and to analyze the current research status of experimental and computational techniques. This thesis analyzes the impact of the irradiation environment on the formation process of interface defects in microelectronic devices. By investigating relevant domestic and international literature and research results, it elaborates, in detail, the damage mechanisms and characteristics of interface defects in microelectronic devices during the annealing process. Finally, based on these research contents, the research methods and current status of the progress in the simulation study of interface defects are discussed. Through the research in this paper, the understanding of the performance damage of microelectronic devices in irradiation environments can be further enhanced, providing a reference for the development of radiation resistant technologies.

## Figures and Tables

**Figure 1 micromachines-16-00434-f001:**
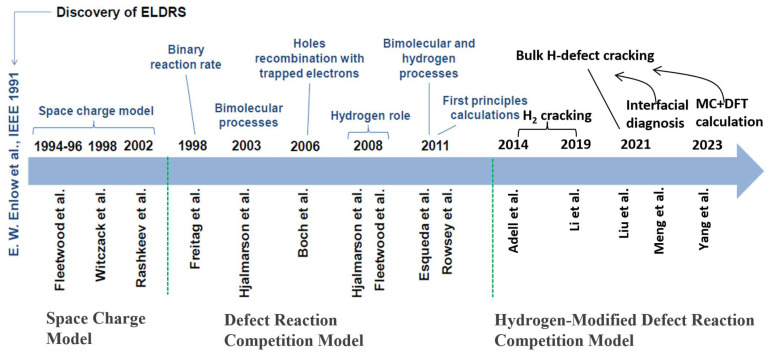
Comprehensive overview of the evolution of models explaining interface traps in MOS and linear bipolar devices [5].

**Figure 2 micromachines-16-00434-f002:**
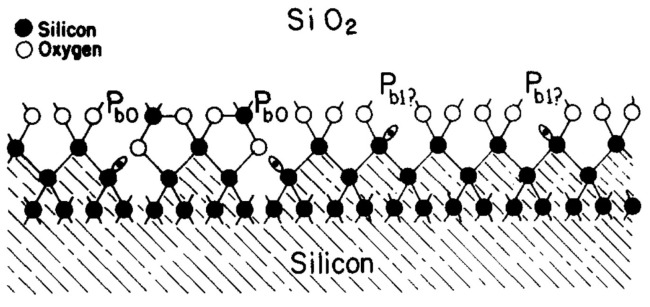
Structural model of the (100) Si-SiO_2_ interface [6].

**Figure 3 micromachines-16-00434-f003:**
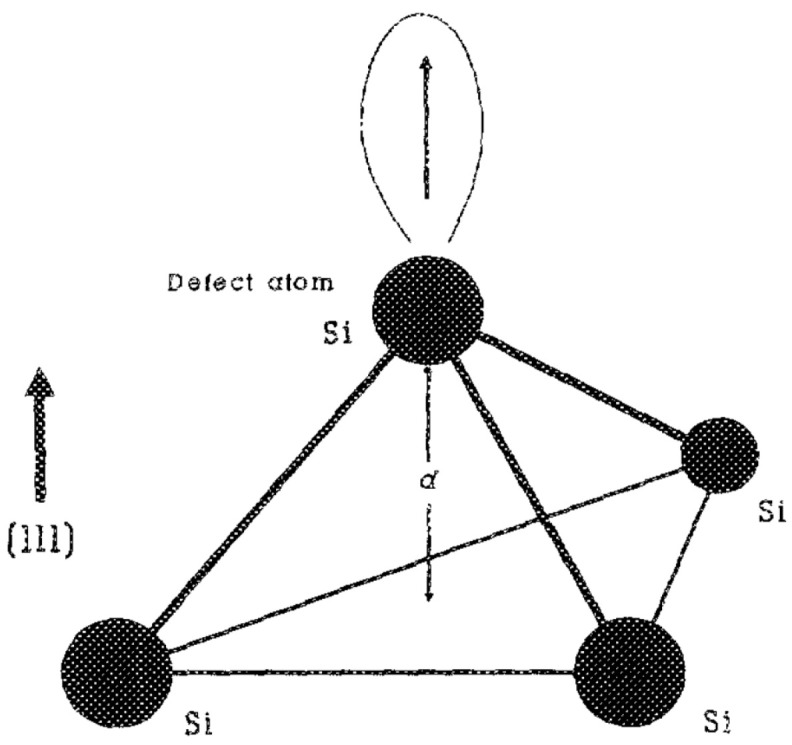
Schematic diagram showing Si tetrahedron containing a dangling bund Si defect, where d is the distance between this defect atom and the plane containing its three nearest neighbors [8].

**Figure 4 micromachines-16-00434-f004:**
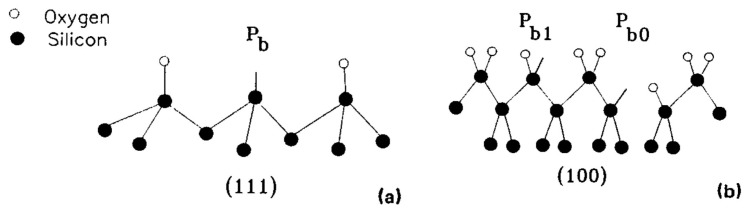
(**a**). Models for P_b_ centers. Dangling bond at the (111) interface. (**b**). Models for P_b_ centers. Poindexter models for P_b_ centers at the (100) interface [9].

**Figure 5 micromachines-16-00434-f005:**
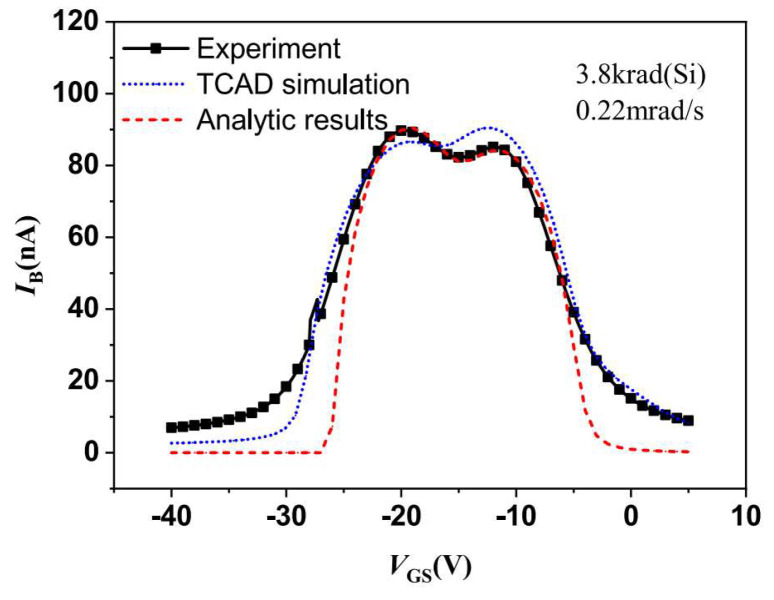
Simulation and experimental results of base current and gate voltage curves [14].

**Figure 6 micromachines-16-00434-f006:**
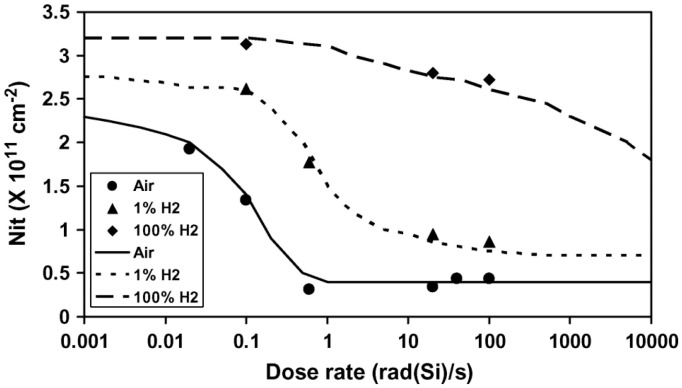
N_it_ versus dose rate for irradiation to 30 krad(Si) for the GLPNPs with p-glass [19].

**Figure 7 micromachines-16-00434-f007:**
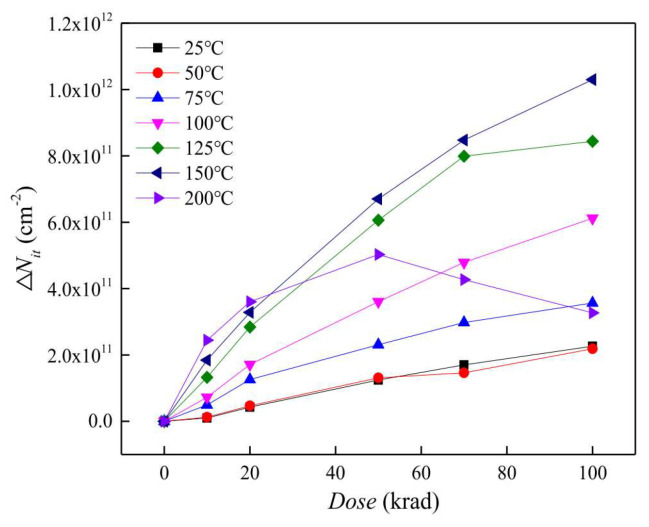
Estimated radiation-induced interface trap buildup versus dose in the oxide for the GLPNP transistors irradiated at different temperatures [24].

**Figure 8 micromachines-16-00434-f008:**
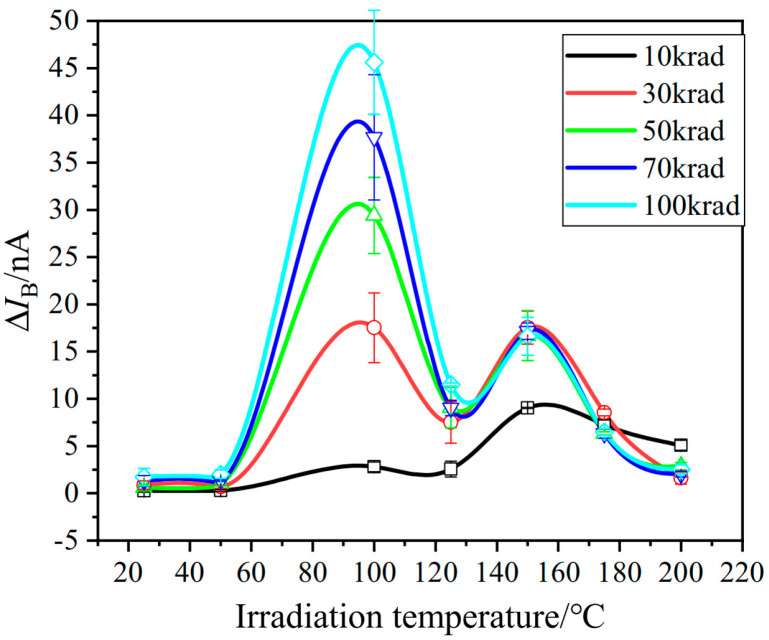
The optimum temperatures at different doses [25].

**Figure 9 micromachines-16-00434-f009:**
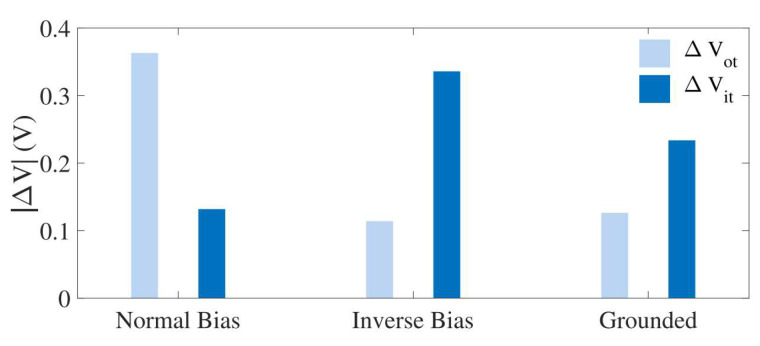
V_ot_ and ΔV_it_ shift after irradiation (400 Mrad (SiO_2_)) for the three different bias conditions [26].

**Figure 10 micromachines-16-00434-f010:**
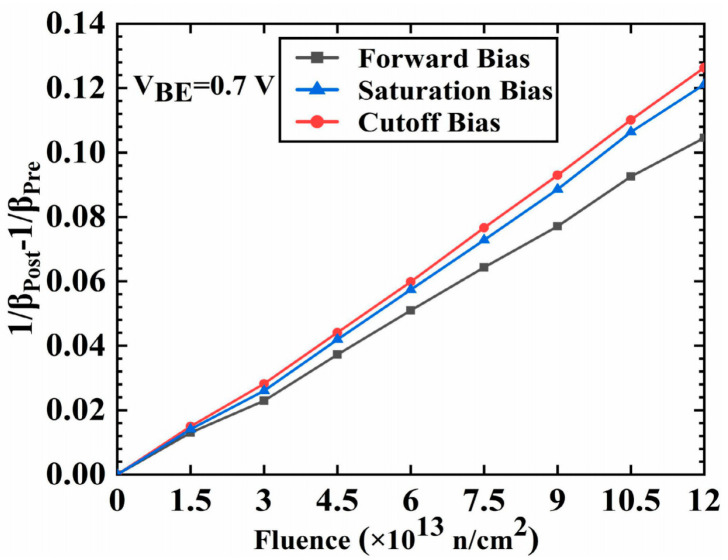
Variation of reciprocal current gain Δ (1/β) with the fluence at a fixed voltage V_BE_ = 0.7 [28].

**Figure 11 micromachines-16-00434-f011:**
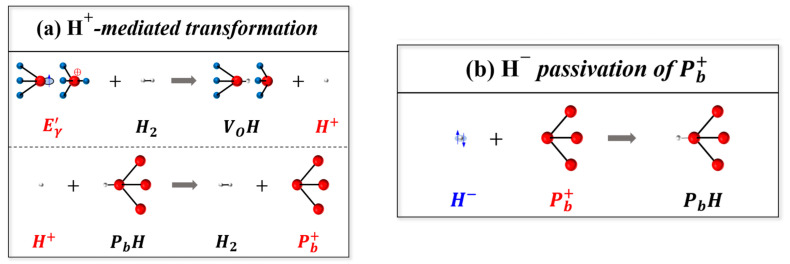
(**a**). Schematic diagrams for H^+^-mediated transformation from $E_{\gamma }^{\prime }$ to P_b_, which can occur at both LT and HT annealing. (**b**). H-induced passivation of P_b_, which only occur at HT annealing. The red, blue, and white balls represent Si, O, and H atoms, respectively.

**Figure 12 micromachines-16-00434-f012:**
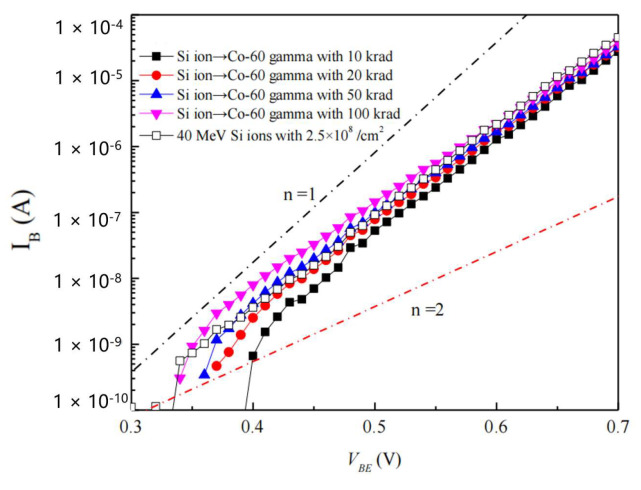
Excess base current versus emitter-base voltage for the NPN 2N2219.

**Figure 13 micromachines-16-00434-f013:**
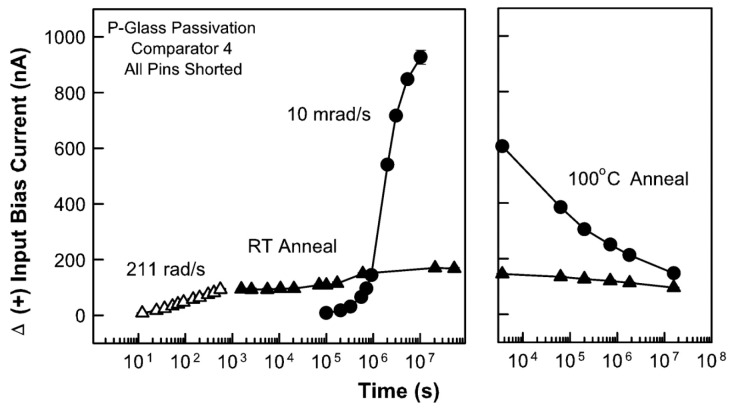
I_B+_ versus time for LM139s passivated with p-glass irradiated at 0.01 and 211 rad(SiO_2_)/s with all pins shorted. Following irradiation, the ICs were annealed at room temperature and elevated temperature with all pins shorted.

**Figure 14 micromachines-16-00434-f014:**
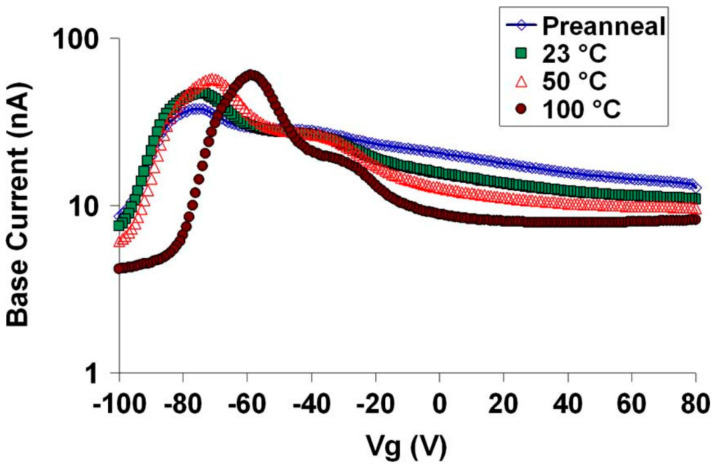
Average gate sweep results after three different temperature anneals among devices with 50 V irradiation biases.

**Figure 15 micromachines-16-00434-f015:**
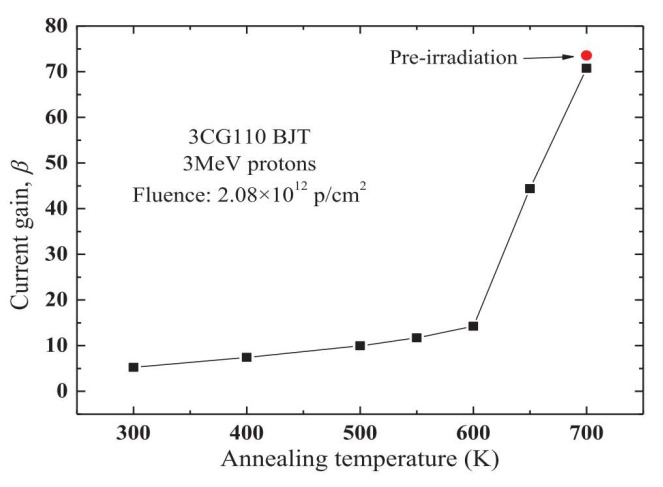
Current gain vs. annealing temperature for the 3CG110 PNP transistor irradiated by 3 MeV protons for a fluence of 2.08 × 10^12^ p/cm^2^.

**Figure 16 micromachines-16-00434-f016:**
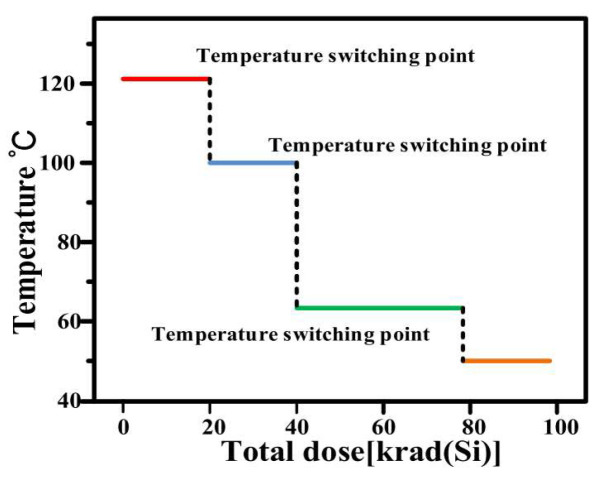
Schematic of the TSI method to characterize ELDRS in linear bipolar technology (after [40]).

**Figure 17 micromachines-16-00434-f017:**
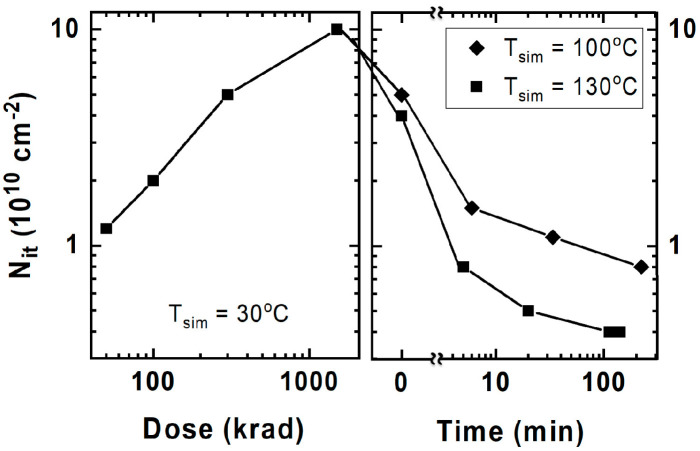
N_it_ as a function of X-ray dose and annealing time.

**Figure 18 micromachines-16-00434-f018:**
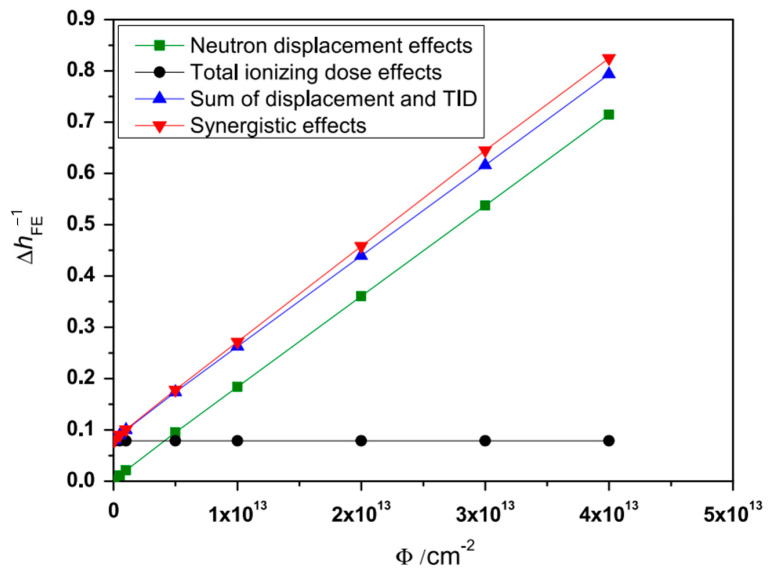
The change in the reciprocal of the common emitter current gain versus neutron fluence in different radiation environments for the lateral PNP bipolar transistor.

**Figure 19 micromachines-16-00434-f019:**
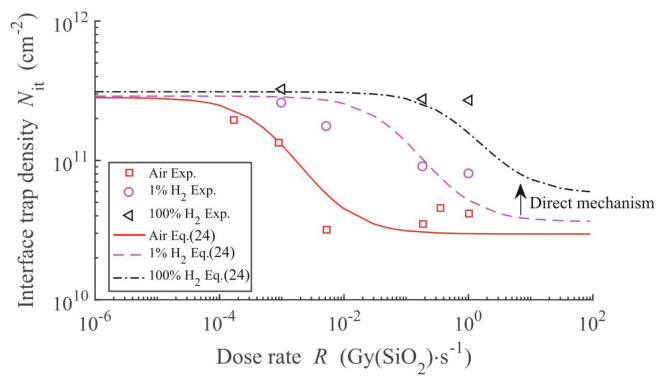
Both the dose rate and hydrogen responses of interface trap density for gate-controlled lateral PNP BJTs after γ-ray exposures up to 300 Gy(SiO_2_) [47].

**Figure 20 micromachines-16-00434-f020:**
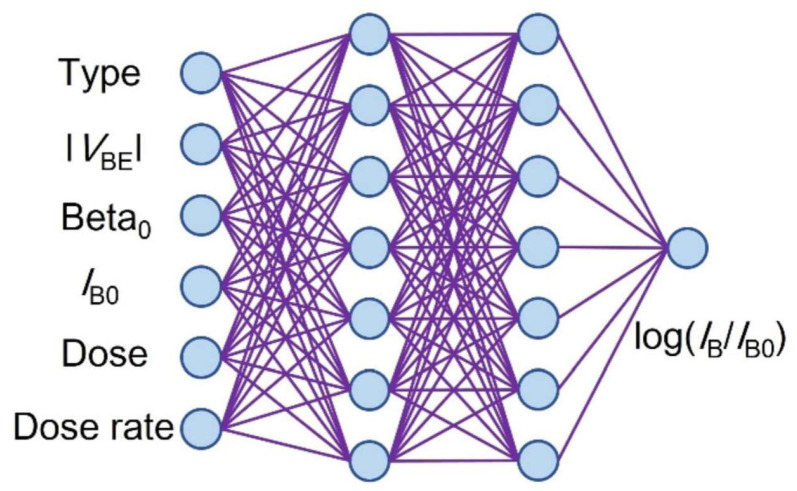
Schematic of the proposed ANN model for our dataset [50].

**Figure 21 micromachines-16-00434-f021:**
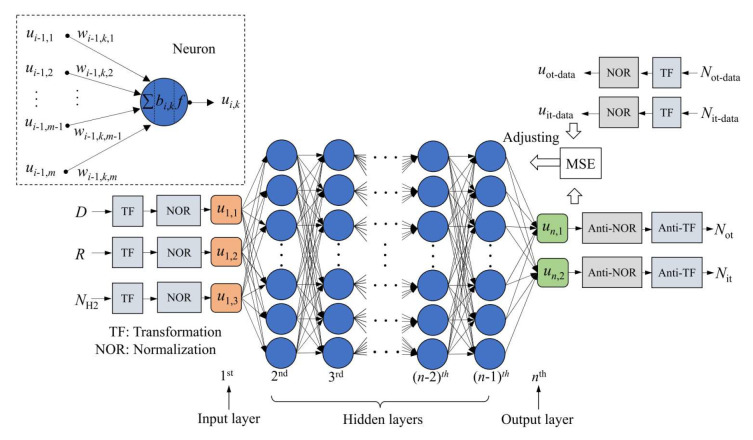
The structure and data flow of the artificial neural network [51].

**Figure 22 micromachines-16-00434-f022:**
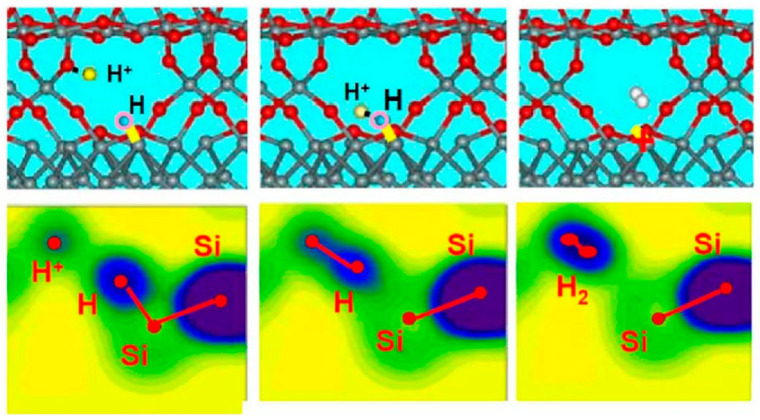
Reaction between H^+^ and an interfacial Si-H bond. The upper panel shows the structure that is simulated via density functional calculations, and the lower panel shows the calculated electron densities around the constituent atoms: Si is shown in gray, O in black, and H in white [55].

**Figure 23 micromachines-16-00434-f023:**
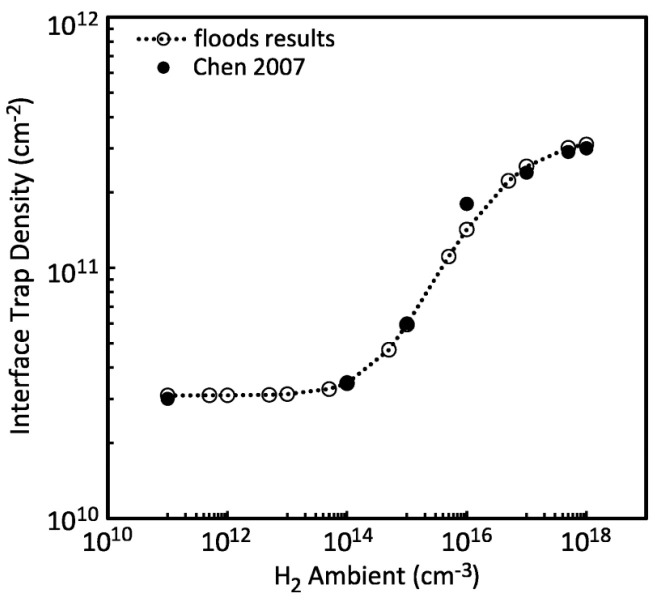
FLOODS results showing a quantitative match to N_it_ data taken by Chen et al. [57].

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
