# Peer review of "Overview of the Properties and Formation Process of Interface Traps in MOS and Linear Bipolar Devices"

_micromachines, 2025, doi:10.3390/mi16040434_

Round 1

Reviewer 1 Report

Comments and Suggestions for Authors

This paper reviews the properties, formation mechanisms, and radiation response of interface traps in MOS and linear bipolar devices. It synthesizes recent research on interface trap transformation, annealing processes, and simulation methods. The study aims to enhance understanding for device design, radiation hardening, and reliability assessment.

To provide readers with a more comprehensive understanding, it would be beneficial to include additional references on interface trap formation and behavior. Specifically, studies on the impact of oxide thickness on TID effects, the temperature dependence of interface traps, and comparative analyses of various annealing techniques would enhance the depth and practical relevance of this research.

  1. Please add reference [R1] in Section 2. The paper [R1] provides valuable insights into the interface trap generation process induced by radiation. [R1] H. J. Barnaby, "Total-ionizing-dose effects in modern CMOS technologies," IEEE Transactions on Nuclear Science, vol. 53, no. 6, pp. 3103-3121, Dec. 2006, doi: 10.1109/TNS.2006.888817. In addition, please add reference [R2] in Section 3. Forming Gas Annealing (FGA), as a recovery technique based on annealing, offers various insights into this topic. [R2] D. H. Wang, S. S. Yoon, J. Y. Ku, D. H. Jung, K. S. Lee, D. Kim, and J. Y. Park, "Low-Temperature Deuterium Annealing for the Recovery of Ionizing Radiation-Induced Damage in MOSFETs," IEEE Transactions on Device and Materials Reliability, vol. 23, no. 2, pp. 297-301, Jun. 2023, doi: 10.1109/TDMR.2023.3275947.
  2. Please include research on the effects of oxide thickness. As the oxide layer becomes thinner, devices tend to exhibit stronger immunity to total ionizing dose (TID) effects.
  3. Please add subheadings to distinguish between MOS and Linear Bipolar Devices.

Author Response

Comments 1:

Please add reference [R1] in Section 2. The paper [R1] provides valuable insights into the interface trap generation process induced by radiation. [R1] H. J. Barnaby, "Total-ionizing-dose effects in modern CMOS technologies," IEEE Transactions on Nuclear Science, vol. 53, no. 6, pp. 3103-3121, Dec. 2006, doi: 10.1109/TNS.2006.888817. In addition, please add reference [R2] in Section 3. Forming Gas Annealing (FGA), as a recovery technique based on annealing, offers various insights into this topic. [R2] D. H. Wang, S. S. Yoon, J. Y. Ku, D. H. Jung, K. S. Lee, D. Kim, and J. Y. Park, "Low-Temperature Deuterium Annealing for the Recovery of Ionizing Radiation-Induced Damage in MOSFETs," IEEE Transactions on Device and Materials Reliability, vol. 23, no. 2, pp. 297-301, Jun. 2023, doi: 10.1109/TDMR.2023.3275947.

Response 1:

In accordance with your specific requirements, we have meticulously cited the relevant references. As indicated in references 16 and 43.

Comments 2:

Please include research on the effects of oxide thickness. As the oxide layer becomes thinner, devices tend to exhibit stronger immunity to total ionizing dose (TID) effects.

Response 2:

Thank you for your valuable comment. We have referenced a study related to the impact of oxide thickness in our literature review, specifically in reference 15.

Titus J L, Wheatley C F, Burton D I, et al. Impact of oxide thickness on SEGR failure in vertical power MOSFETs; development of a semi-empirical expression[J]. IEEE Transactions on Nuclear Science, 1995, 42(6): 1928-1934.

Comments 3:

Please add subheadings to distinguish between MOS and Linear Bipolar Devices.

Response 3:

Thank you for your valuable comment. I have added appropriate subheadings in the paper to clearly distinguish between MOS and Linear Bipolar Devices, which will enhance the readability and organization of the content.

Reviewer 2 Report

Comments and Suggestions for Authors
  1. This review covers radiation impacts, annealing, and annealing simulations, but each section mechanically lists studies chronologically, merely describing each work without analysis or connection. Recent progress comparisons would be more insightful than timelines. Also, the author should add personal viewpoints and critiques to stand out from other reviews.
  2. Figures and captions shouldn't cross pages, e.g., Figure 3 and Figure 14.
  3. One large figure can include multiple related smaller ones with labels (like Figure 1a, 1b, 1c), instead of each large figure having only one small one.
  4. An overall picture summarizing the current understanding of interface trap in MOS and linear bipolar devices would be beneficial in the first part.

5 The review focuses too much on recent progress, and needs more analysis of challenges and critical difficulties.

Author Response

Comments 1:

This review covers radiation impacts, annealing, and annealing simulations, but each section mechanically lists studies chronologically, merely describing each work without analysis or connection. Recent progress comparisons would be more insightful than timelines. Also, the author should add personal viewpoints and critiques to stand out from other reviews.

Response 1: Thank you for the valuable feedback. In response, I will restructure the review to focus on comparing recent progress and insights across studies rather than presenting a chronological list. I will also incorporate personal viewpoints and critiques to provide a more analytical perspective, highlighting key advancements and identifying gaps in the current understanding.

Comments 2:

Figures and captions shouldn't cross pages, e.g., Figure 3 and Figure 14.

Response 2: Thank you for pointing this out. I will ensure that figures and their corresponding captions are placed on the same page to maintain clarity and readability.

Comments 3:

One large figure can include multiple related smaller ones with labels (like Figure 1a, 1b, 1c), instead of each large figure having only one small one.

Response 3: Thank you for the suggestion. I will consolidate related smaller figures into a single large figure with labels (e.g., Figure 4a, 4b,) to improve clarity and save space.

Comments 4: An overall picture summarizing the current understanding of interface trap in MOS and linear bipolar devices would be beneficial in the first part.

Response 4: Thank you for the suggestion. I will include a comprehensive summary figure in the first part of the review to provide an overall picture of the current understanding of interface traps in MOS and linear bipolar devices. This figure will highlight key mechanisms, models, and recent advancements, offering a clear and concise overview before delving into detailed discussions.

Comments 5: The review focuses too much on recent progress, and needs more analysis of challenges and critical difficulties.

Response 5: Thank you for the feedback. I will balance the review by incorporating a deeper analysis of the challenges and critical difficulties in the field. This will include discussing unresolved issues, limitations of current models, and potential future directions, providing a more comprehensive and critical perspective alongside recent progress.

Round 2

Reviewer 2 Report

Comments and Suggestions for Authors

The author has made many revisions to the manuscript, and the quality has improved. But some problems remain: the figures (e.g., Figures 2, 3, and 4) have obvious screenshot marks and poor quality. The fonts in each figure are inconsistent, and it's unclear if they're from other literature as there are no citations or copyright requests. Also, Figures 4a and 4b should be combined into one.